# How to Address Consumers’ Concerns and Information Needs about Emerging Chemical and Microbial Contaminants in Drinking Water; The Case of GenX in The Netherlands

**DOI:** 10.3390/ijerph182010615

**Published:** 2021-10-11

**Authors:** Liesbeth Claassen, Julia Hartmann, Susanne Wuijts

**Affiliations:** 1National Institute for Public Health and the Environment (RIVM), P.O. Box 1, 3720 BA Bilthoven, The Netherlands; liesbeth.claassen@rivm.nl (L.C.); susanne.wuijts@rivm.nl (S.W.); 2Faculty of Civil Engineering and Geosciences, Delft University of Technology, P.O. Box 5048, 2600 GA Delft, The Netherlands; 3Utrecht Centre for Water, Oceans and Sustainability Law, Utrecht University, Newtonlaan 231, 3584 BH Utrecht, The Netherlands

**Keywords:** drinking water, emerging contaminants, mental models, risk perception, risk communication

## Abstract

The perceived safety of tap water is an important condition for consumers to drink it. Therefore, addressing consumers’ concerns should be included in the roadmap towards the UN SDG 6 on safe drinking water for all. This paper studies consumers’ information needs regarding emerging contaminants in drinking water using a mental model approach for the development of targeted risk communication. As most consumers expect safe drinking water, free of contamination, communication on emerging contaminants may increase concerns. Here, we showed that communication strategies better tailored to consumers’ information needs result in smaller increases in risk perception compared with existing strategies.

## 1. Introduction

Safe tap water is crucial in maintaining public health as it is used for drinking water purposes, personal hygiene and food preparation [1,2]. According to the World Health Organization (WHO), tap water is considered safe when it does not represent any significant risk to health over a lifetime of consumption, including different sensitivities that may occur between life stages [3]. In addition to this physical aspect of safe drinking water, the safety of tap water as perceived by consumers is also crucial because consumers who perceive tap water as unsafe will search for alternatives to drink, e.g., bottled water or sodas, which is undesirable from a sustainability and health perspective [4,5,6]. For this reason, addressing consumers’ concerns should be included in the roadmap towards the United Nations Sustainable Development Goal number 6 on safe drinking water for all [7]. Initiatives such as the Right2Water initiative demonstrate that consumers also acknowledge the importance of safe drinking water [8]. Protecting both the physical and perceived safety of tap water throughout the world is thus vital in safeguarding public health. This paper aims to contribute to this protective role by developing targeted risk communication regarding contaminants in tap water, based on consumers’ information needs.

Given that the focus here is on the exposure to tap water through drinking, tap water will be referred to as drinking water. In Europe, approximately 40 per cent of drinking water from the tap is produced from groundwater and 60 per cent from surface water [9]. Both groundwater [10,11] and surface water [12] are susceptible to chemical and microbial contamination from various anthropogenic activities, such as agriculture [13,14] and wastewater discharges from municipalities and industry [15,16,17].

In addition to well-known drinking water contaminants, several (often unregulated) emerging chemical compounds have been detected in drinking water and its resources over the past decades, posing a new or increased threat to public health [17,18,19,20,21,22,23]. These chemicals are referred to as *emerging contaminants, emerging pollutants* or *contaminants of emerging concern* [24]. Recent examples are perfluoroalkyl and polyfluoroalkyl substances (PFAS) [25], 1,4-dioxane [21] and quaternary phosphonium compounds [20]. In addition to emerging chemical contaminants, emerging contaminants of a microbial nature have also been identified in drinking water in recent years [26,27].

The issue of emerging chemical and microbial contaminants in drinking water resources is expected to increase due to: (1) demographic developments (e.g., increased consumption of pharmaceuticals by a growing and ageing population [28,29]); (2) societal changes (e.g., growing industrial activities [30,31]; (3) technological improvements (e.g., the increasing sensitivity of analytical techniques [17]); (4) regulatory changes (e.g., use of even more hazardous alternatives as a result of phasing out of specific contaminants, referred to as *regrettable substitution* [32]); and (5) climate change (e.g., discharges of untreated wastewater due to sewage overflows during heavy rain events [33,34]). Given these developments, providing consumers with safe tap water now and in the future remains a challenge.

Following the definition in the WHO Guidelines for Drinking-Water Quality, safe drinking water is not necessarily free of contaminants. Still, it must be free of levels of contaminants that pose a significant threat to humans [3]. This is based on the notion that a contaminant may have hazardous properties, but the risk it poses to human health depends on the level of exposure. To this end, so-called health-based guideline values (HBGV) in drinking water are calculated, representing the concentration in drinking water that does not adversely affect human health even over a lifetime consumption of that drinking water. 

Risk assessment of chemicals in drinking water differs for threshold and non-threshold chemicals [3]. For threshold chemicals, it is assumed that a level of exposure exists below which no adverse health effects occur. This level is typically based on a no-observed-effect level found in animal toxicity testing, which is then extrapolated to humans using uncertainty factors (UF) [3,35,36]. Uncertainty factors can range from 10 to 10,000 based on the reliability and relatability to humans of the available data. The obtained value is multiplied by the average weight of a human and the relative importance of the exposure to the chemical via drinking water compared to other routes (e.g., air or food) [37]. Finally, the estimated daily intake of drinking water is taken into consideration (e.g., 2 L) [3,35,36]. For genotoxic carcinogens, so-called non-threshold chemicals, it is believed that exposure to one additional molecule can cause cancer by inducing DNA mutations [38]. Therefore, acceptable levels in drinking water are based on the concentration leading to a theoretical acceptable excess lifetime cancer risk (e.g., 1 excess case of cancer per 100,000 people according to the WHO [3]). 

When it comes to emerging contaminants, toxicity data might be unreliable or insufficient and other risk assessment approaches need to be used to define HBGVs, such as the threshold of toxicological concern (TTC) approach [37,39,40,41]. The TTC is a level of daily intake that poses no significant risk to human health. This is based on the notion that contaminants with similar structures have similar toxic properties. This maximum level of daily intake per person per day can be combined with the daily intake of drinking water to calculate an HBGV. The use of the TTC approach and related approaches to derive HBGVs has been explained in depth elsewhere [37,39,42,43]. HBGVs may differ between countries in the case of risk assessment based on contaminant specific toxicity data (e.g., use of different standard body weights or different exposure allocations to drinking water) as well as when using the TTC approach (use of different threshold values for different classes in the TTC approach) [22,44,45]. The uncertainty associated with the risk assessment of emerging drinking water contaminants, as well as the international differences in what is considered safe drinking water, can be a challenge in risk communication [22].

With respect to the quality of their drinking water, consumers rely on science to identify and monitor hazards, determine health risks and inform them about potential health threats, and on policymakers to regulate these threats, by enforcing strict safety levels. Frequently, consumers are confronted with information stating that a hazardous chemical or microbial contaminant has been detected in their drinking water but that this presence does not constitute a risk to public health because the doses do not exceed the safety levels. While experts may be familiar with such statements, public understanding of these risk statements generally does not correspond with the scientific interpretation [46,47].

First, for consumers, the difference between ‘hazard’ and ‘risk’ is not always clear; they tend to consider all hazards as risks. This is partly due to the use of ‘risk’ instead of ‘hazard’ in risk communication [48]. Often, there is also a mismatch between the intended (scientific) meaning and consumers’ interpretation of terminology describing the hazardous qualities of a contaminant [47,49,50]. For example, ‘possibly carcinogenic’ is often understood by the public as ‘likely to cause cancer’ [47]. A second potential mismatch is that references to safe exposure levels in risk information are mostly provided in scientific terms such as ‘acceptable daily intake’ without translating these into familiar language and meaningful amounts for consumers. Third, although many consumers can distinguish the conceptual difference between hazard and risk [47,48,51], Dutch citizens generally think their drinking tap water is very safe [52] and thus free of contaminants. The mere presence of pathogens or chemicals with pathogenic or toxicological properties may violate that belief. Moreover, consumers may doubt that there is such a thing as a safe amount of a contaminant that can cause serious harm in higher doses. Fourth, to interpret the unfamiliar information, non-experts rely on beliefs associated with known hazards, based on perceived similarity of characteristics [46]. These associations are strengthened by personal experience with and knowledge of the known hazard, and information provided by media. For example, people tend to associate the effects of phthalates on unborn children with the effect of the drug Softenon (thalidomide) [46].

The aforementioned communication mismatches can create misunderstandings, which may result in unintended and undesirable consumer behaviour, such as avoiding drinking water from the tap [6], particularly when hazardous contaminants have been detected in their drinking water. For a more effective risk communication strategy that bridges the gap between experts using complex scientific and technical knowledge and terminology, and consumers who often base their risk judgments on prior beliefs, lay interpretations of terminology and proximity of the risk are required.

A better understanding of consumers’ mental models underlying their beliefs and behavioural decisions is a good starting point for developing risk communication materials [53]. The differences between consumers’ mental models and experts’ representations of the risk assessment process should provide clear insights into the consumers’ information needs. When consumers receive information that fits their mental models and adequately addresses their information needs, this improves their understanding of the risk assessment and management processes so that they are able to make informed and independent judgments about the hazards that they face and about the adequacy of mitigation policies [53].

An often used medium to clarify data and explain concepts in science communication is the use of visual illustrations or embellished infographics. Such infographics can summarise, reduce and simplify information, highlight the most important aspects and convey the complexity of interrelated processes, such as the risk assessment process. There is some evidence that well-designed visual materials offer an effective means of communicating risk numbers [54]. However, evidence for the effectiveness of infographics to explain concepts and processes is scarce [55,56,57]. Some studies show that infographics illustrating concepts are easier to comprehend and remember than text alone [55,57]. Fandel, Breshears [56] and Guo, Zhang [57] also point to various aspects of infographic design that can have a considerable impact on comprehension, such as the use of ambiguous or irrelevant illustrations that are not clearly linked to the corresponding text. Moreover, comprehension of an infographic often varies widely within a population, revealing that many people have great difficulty in understanding the meaning.

In the present study, we tested the effectiveness of textual and visual risk communication materials on emerging contaminants, both chemical and microbial, in drinking water. Based on mental models research, we redrafted an existing web text on the presence and health risk of the chemical GenX, a recently identified PFAS [58] in drinking water, and developed an accompanying infographic explaining the general risk assessment process. GenX substances are used in the production of fluoro-polymers. Fluoro-polymers have many applications, such as in coatings. In June 2019, the European Chemicals Agency unanimously decided that GenX substances are substances of very high concern based on their mutagenic, carcinogenic, reproductive toxic and bioaccumulative properties, as well as their persistence in the environment. Specific information on GenX presence in drinking water in the Netherlands as well as the Dutch health-based guideline value is included in the Appendix A. We performed an experiment to assess the effects of alternative risk information on judgments and decision making.

## 2. Materials and Methods

### 2.1. Development of Materials

To identify relevant consumer information needs and to construct effective communication strategies about risks from emerging chemical and microbial contaminants in drinking water, we followed the mental models approach to risk communication [53]. The approach used in this study is depicted in Figure 1.

First, to construct an expert/stakeholder mental model, the project team, consisting of scientists selected for their expertise on chemical and microbial drinking water quality, risk assessment and risk communication (for details, see acknowledgements), mapped relevant professional stakeholders from Dutch drinking water companies, industry and responsible authorities (hereafter referred to as ‘stakeholders’). These stakeholders were invited to participate in two workshops on November 9th, 2017 and November 6th, 2018, organised by the project team.

The aim of the workshops was to describe stakeholders’ perspectives on safe drinking water. These perspectives were used as an indication of what consumers need to know about safe drinking water. In Workshop 1, stakeholders’ perspectives were gathered using key questions on: (1) the quality and safety of Dutch drinking water (including threats, treatment, pollution sources and risk factors); (2) stakeholders’ concerns regarding drinking water safety in the Netherlands; (3) potential mitigation actions; and (4) risk communication strategies. The project team used the information gathered in Workshop 1 and complemented it with the scientific perspective on drinking water safety to construct a comprehensive mental model. Stakeholders reviewed the expert/stakeholder model during Workshop 2.

Subsequently, a mental model of consumers’ perspectives was constructed to define what consumers already know about drinking water safety and what their information needs are. The consumers’ mental model was based on: (1) previous research [52,59]; (2) short interviews (*n* = 15) on perceptions of drinking water quality in a convenient sample (visitors of a public science day); and (3) more in-depth telephone interviews (*n* =13) with members from a consumer panel of a Dutch internet research agency (Flycatcher Internet Research). This research agency has its own panel consisting of more than 10,000 members representative of the Dutch population (in composition of age/gender/education and distribution over different parts of the country) and is in possession of a quality label for market, opinion and social research (ISO 20252) and is certified according to the environmental standard (ISO 14001). The telephone interviews focused on what people perceived as important threats to the quality of drinking water, what they knew about water management and safety limits and their information needs concerning drinking water quality. 

The expert/stakeholders’ mental model was compared with the consumers’ mental model to identify relevant knowledge gaps, common lay beliefs and questions regarding drinking water safety. We found that consumers generally acknowledge industrial wastewater discharges and runoff from agricultural land as significant threats to drinking water safety; some consumers also specifically mentioned the presence of GenX in drinking water. However, others thought that safe drinking water is entirely free from contaminants. Most consumers did not know how the safety of drinking water is monitored. Finally, the meaning of a safety limit (HBGV) was not clearly understood (see also Table 1).

Several of the knowledge gaps, lay beliefs and questions (see italics in Table 1) identified, were then addressed in the revision of a published text from the website of the Dutch National Institute for Public Health and the Environment (RIVM) (https://www.rivm.nl/genx/drinkwater (last accessed on 4 October 2021), in Dutch). The chosen text addresses the recent public and scientific attention for the presence and health risk of GenX, a recently identified PFAS [58], in drinking water, and informed consumers about the risk assessment. A general supporting infographic was designed (following recommendations of Fandel, Breshears [56]) to clarify the risk assessment process for chemical and microbial contaminants in drinking water (addressing the question in bold in Table 1).

A pilot test was then conducted among 21 members from the Flycatcher Internet Research panel, who commented on the clarity of the materials and survey items (see measurements in paragraph 2.4 and Figure 1), before the actual survey was conducted. Based on the responses from the pilot test, several adjustments in formatting and wording were made.

The final version of the original text, the revised web text and the infographic (all three are available in Dutch and English) were then tested in an online experiment among 510 participants (materials are included in Appendix A) as indicated by Figure 1. 

### 2.2. The Online Experiment

An online survey was used for data collection. In May 2020, participants were invited by e-mail to participate in a survey on drinking water safety by clicking on a unique hyperlink in the invitation. The survey consisted of three consecutive parts. The first part contained questions concerning drinking water safety, trust in information and regulation (see pre-text measurement described in Section 2.4.1). In the second part, participants first received basic information about the presence of GenX in drinking water and were then randomly allocated across the three information conditions (all texts are included in the Appendix A (in Dutch and English)):Condition 1: Existing website text on GenX;Condition 2: Alternative website text on GenX;Condition 3: Alternative website text on GenX and risk assessment infographic.

In the third section, participants were presented with questions about how they evaluated the information, concerns about hazards after reading the text, acceptance of risk management and drinking water use (see Section 2.4.2). 

### 2.3. Participants

We invited participants through Flycatcher Internet Research. The members in their panel can state their voluntary and active participation in online research through a double-active opt-in procedure. By participating in a study, members can earn points that can be exchanged for gift cards. A total of 510 members participated in the survey study (response rate 67%), of whom 259 were residents of the South Western (S-W) provinces and 251 of other parts of the Netherlands. The S-W provinces of The Netherlands are relevant here, as GenX was found in the drinking water from this region [58]. Both samples were representative of the Dutch general population in 2019 with respect to age, gender and education level.

### 2.4. Measurements

#### 2.4.1. Pre-Text Measurement Variables

Participants of the survey were asked to rate the *quality of Dutch drinking water* (‘The drinking water from my tap is…’) on four 5-point bipolar semantic differential scales, each representing a specific quality adjective (‘tasty’—‘not tasty’, ‘unsafe’—‘safe’, ‘healthy’—‘unhealthy’ and ‘bad’—‘good’). The internal consistency of this scale (with negative scales reverse coded) was satisfactory (α = 0.77).

Next, to assess their *knowledge about drinking water safety*, participants were asked to state to what extent (on a scale from 1= ‘definitely false’ to 5 = ‘definitely true’) they thought the following five statements, formulated in line with expert opinions, were true: ‘Drinking water companies check the drinking water for harmful substances on a daily basis’, ‘There are harmful substances in the drinking water’, ‘You can ingest a little bit of a harmful substance every day without ever getting ill’, ‘Researchers can determine the amount of a harmful substance for which drinking water is safe’, and ‘If the drinking water is safe, anyone can drink it daily’. In addition to these knowledge questions, one specific lay belief which is not in line with expert opinions was assessed: ‘Drinking water is only safe if it contains no harmful substances’. The response to this last statement was reversely coded, following which a knowledge sum score over the six items was calculated.

Subsequently, participants were asked to what extent (on a scale from 1 = ‘not at all’ to 5 = ‘a lot’) they had *concerns* about the four following potential contaminants in their drinking water: carcinogenic substances, endocrine disruptors, bacteria and other microorganisms and calcium. These items are analysed separately.

In addition, we assessed *trust in risk management* by asking to what extent (on a scale from 1 = ‘not at all’, to 5 = ‘a lot’) participants trusted the information of the Dutch National Institute for Public Health and the Environment (RIVM), the information of drinking water companies and laws and regulation concerning drinking water. These three items formed an internally consistent scale (α = 0.86).

#### 2.4.2. Post-Text Measurement Variables

After participants went through the provided information, we asked them to indicate (on 5-point bipolar semantic differential scales) whether they thought the information was ‘difficult’—‘easy to understand’, ‘complete’—‘incomplete’ (reverse coded), ‘not trustworthy’—‘trustworthy’, ‘new’—‘known’, ‘clear’—‘vague’ (reverse coded) and ‘inconsistent’—‘consistent’. They were also asked to clarify their responses (open question). Except for newness of information, the responses formed an internally consistent *evaluation of information* scale (α = 0.82).

Next, we reassessed their concerns about contaminants in drinking water (carcinogenic substances, endocrine disruptors, bacteria and other microorganisms and calcium) using the pre-text measurement variables.

In addition, we asked participants questions about their *perceptions of GenX risk* (on 5-point rating scales; 1 = ‘certainly not’, 5 = ‘certainly’): ‘I ingest GenX via drinking water’, ‘You can develop cancer due to the amount of GenX in the drinking water’, ‘I think the information is reassuring’ (reverse coded), ‘I think my drinking water is safe to drink’ (reverse coded), ‘I can get ill due to the amount of GenX in my drinking water’, ‘I think the risk of GenX in drinking water is high’, ‘I worry about the harmful effects for children’ (internal consistency α = 0.85).

*Acceptance of norms* and regulation was assessed by asking participants whether they thought (on 5-point rating scales; (1 = ‘certainly not’, 5 = ‘certainly’)): ‘The norms for GenX in drinking water are strict enough’, ‘A small amount of GenX in drinking water is acceptable’, ‘As long as the amount of GenX in drinking water stays within the safe norm, it is all right’, ‘The risk of GenX in drinking water is too high’ (reverse coded) (internal consistency α = 0.89).

And finally, we assessed participants’ self-reported *restricted water use* with four items (on 5-point rating scales; 1 = ‘certainly not’, 5 = ‘certainly’): ’I drink it without limitations’ (reverse coded), ‘I prefer bottled water’, ‘I only drink the water after boiling (e.g., for tea)’, ‘I filter the water before I drink it’. A sum score was calculated, ranging from 4 (not restricted) to 20 (very restricted).

### 2.5. Statistical Analyses

After checking for socio-demographic differences between information conditions and regions, using χ2 tests for gender and education level and analyses of variances (ANOVAs) for age, we performed t-tests to assess the regional differences in pre-text measurement variables. Next, we calculated Pearson’s correlations between variables and performed ANOVAs to test for informational effects of the alternative text, with and without an infographic, on the post-text measurement variables with region and condition as fixed factors.

## 3. Results

### 3.1. Sample

Of the 767 panel members recruited, 510 participated in the study (response rate 67%) (see Figure 1). Most participants completed the survey within 10 min (95% CI (7.6–9.8 min)). The demographic characteristics of the participants are presented in Table 2. There were no significant differences in demographic characteristics between the two regions.

Table 3 presents the pre-text evaluations of the study sample. No differences were found between regions with respect to the evaluation of the quality of drinking water, knowledge about drinking water safety and concerns. However, there was a small but significant difference in trust in risk management; in the S-W region, participants demonstrated less trust compared with the other regions (t(508) = −2.202, *p* = 0.028).

### 3.2. Associations between Variables

Pearson’s correlations between variables are presented in Table 4. There was a relatively strong positive association between the pre-text measurement variables trust in risk management and the evaluation of the quality of drinking water. Both variables were also positively correlated with the evaluation of information and acceptance of norms and negatively associated with post-text concerns and perceptions of GenX risks and restricted water use. All associations between knowledge about safety and the other pre-text and post-text measurement variables were either non-significant or low (r < 25). Table 4 also shows relatively strong associations between the post-text measurement variables. As expected, lower concerns and perceptions of GenX corresponded with greater acceptance of norms and fewer self-reported restrictions in water use. Notably, more positive evaluations of information corresponded with lower concerns and perceptions of GenX and greater acceptance of fewer restrictions in water use.

### 3.3. Effects of Web Text and Infographic

Table 5 shows the results of an interaction effect of both condition and region on the evaluation of information. There was no significant difference between the conditions with and without the infographic. The open comments in response to the infographic varied strongly: it was perceived as complicated and difficult to understand by some, whereas others felt it clarified the process.

For post-text concerns, we found effects within subjects. After going through the provided information, participants’ concerns increased with respect to carcinogenic substances (F(1,504) = 268.904, *p* < 0.001), endocrine disruptors (F(1,504)= 86.309, *p* < 0.001) and microorganisms (F(1,504)= 20.449, *p* < 0.001) in drinking water, while concerns with respect to calcium in drinking water decreased (F(1,504) = 8.686, *p* = 0.003). There was a between-subjects interaction effect of pre-text concerns and condition on post-text concerns: participants in the conditions with the alternative text showed smaller increases in concerns about carcinogenic substances (F(2,504)= 8.359, *p* < 0.001) and endocrine disruptors (F(2,504) = 3.589, *p* = 0.028) compared with participants in the condition with the existing web text (but not for microorganisms and calcium). This interaction effect was independent of the region. There was no significant difference between alternative conditions with and without the infographic.

Perception of GenX risks was higher in the S-W region compared with the other regions (F(1,504)= 5.119, *p* = 0.024) and lower in the conditions with the alternative text compared with the existing web text (F(2,504)= 3.971, *p* = 0.019). There was no significant difference between the alternative conditions with and without the infographic.

In the S-W region, acceptance of norms was lower (F(1,504) = 5.368, *p* = 0.021)) and restricted water use was higher compared with the other regions (F(1,504) = 5.914, *p* = 0.015)). There was no significant effect of conditions on acceptance of norms and drinking water use.

## 4. Discussion

In this study, mental models-based risk information on microbiological and chemical hazards in drinking water such as GenX was developed and an experiment was carried out to assess the effects of providing this information on judgments and decision making by consumers. To this end, the expert/stakeholders’ mental model was compared with the consumers’ mental model to identify their concerns and information needs (e.g., the meaning of a safety limit (HBGV)). 

The results show that in the context of this study, risk communication materials that are developed with the information needs from consumers in mind, are appreciated more and result in smaller increases in risk perception compared with existing materials. In this section, we will reflect on these results, their relevance in the international context and the strengths and limitations of the methodology applied, and explore some avenues for future research.

### 4.1. Reflection on Results

In the experiment, participants were assigned to one of three alternative information packages: the original text from the website on GenX, a modified text that aimed to meet information needs previously identified among consumers or the modified text complemented with an infographic on the process of risk assessment for emerging chemical and microbiological contaminants in drinking water. The results showed that the modified text was better understood and appreciated in the region with higher GenX emission, indicating that the modifications addressed consumers’ information needs better. Moreover, the modified text prompted less apprehension about GenX risk compared with the existing website text. As these factors were also associated with greater acceptance of GenX norms and regulation, and fewer self-reported restrictions in water use, the modified mental models-based information material delivered promising outcomes. Using a mental models approach to ensure that risk communication materials take consumers’ information needs into account has also been found effective in improving risk comprehension and in changing attitudes and behaviour towards risk in other studies (see review by Boase et al. [60]). In a Mexican study, for example, a comic book explaining carbon monoxide (CO) poisoning risks to residents reduced misunderstanding about CO risk and increased the willingness to use CO alarms [61].

The use of the infographic explaining the risk assessment process did not result in improved understanding and appreciation of the modified text. Nor did it affect risk apprehensions, although it should be noted that it did not have adverse effects either. There are many factors that may explain the lack of effect. Meta-analyses of studies using illustrations to improve understanding showed that although graphics accompanying texts can facilitate readers’ comprehension [57], many features can hamper comprehension. Graphs, which require more investment from the reader, often yield less robust effects. In particular, the density and variety of visuals, the intricacy of individual visual representations and the spatial and semantic integration of text and visuals demand cognitive effort from readers. In our study, the infographic was indeed perceived as rather complex by some, possibly because the infographic offered a lot of information to digest or did not clarify the link with the provided text on GenX satisfactorily, since the infographic focused on drinking water and emerging contaminants *in general* and was not tailored to GenX.

We were specifically interested in the responses of participants in the S-W region. The S-W region harbours a few large chemical industries, and over the past few years, reports have been published on the presence of PFAS in the drinking water in this region. The first reports were about possibly contaminated groundwater, which had been used for the production of drinking water. The groundwater was found to be polluted by PFOA due to industrial wastewater discharges and subsequent infiltration into groundwater in the period of 1970–2012 [22,62,63]. Then, in 2017, GenX, which has been used as an alternative for PFOA by the chemical industry since 2012, was also detected in the drinking water in this region [58,64]. These results require risk communication specifically targeted to consumers in this region, for whom it is most relevant [65]. It may not be surprising that the participants in this region were more vigilant with respect to drinking water safety. They demonstrated less trust in risk management, were less satisfied with GenX norms and regulations and also reported more restrictions in water use compared with the other regions. However, they were also more susceptible to this targeted communication material. As noted above, they evaluated the modified information more positively compared with the existing text.

Before reading the information, participants in both regions were more concerned about the water’s hardness than about chemical and microbial contaminants. It is likely that they were previously unaware of the presence of hazardous contaminants in their drinking water. The provision of information on the presence of GenX resulted in increases in concerns about not only carcinogenic substances but also about endocrine disruptors and microbiological contaminants (interestingly, however, concerns about the hardness decreased). These increases in concerns may trigger unintended and undesirable consumer behaviour, such as avoiding drinking water from the tap. Nevertheless, keeping consumers in the dark about the presence of contaminants is not a viable option. The real challenge is to adequately inform people about the presence of contaminants to enable them to make informed decisions that they face about the adequacy of norms [53].

### 4.2. International Context

In a previous study by Hartmann, van der Aa [22], several risk communication strategies used in Minnesota (USA), Germany, Switzerland and the Netherlands on another PFAS (perfluorooctanoic acid) in drinking water were analysed. Hartmann, van der Aa [22] concluded that timely communication resulted in lower concern among the public. However, in all studied countries, mostly one-way technical communication strategies [66] were used to inform, assure and possibly change the behaviour of consumers, by conveying scientifically determined risk [22]. With the present study in mind, we suggest that taking concerns and questions of consumers on GenX in drinking water into account could have resulted in more adequate risk communication in the cases studied by Hartmann, van der Aa [22]. Given that the cases studied by Hartmann, van der Aa [22] also consider a PFAS in drinking water, similar information needs to the ones identified in this study are expected (see Table 1). For instance: ‘What does it mean when a contaminant is present in drinking water under the (safety) limit?’. Such an approach may offer valuable input for developing risk communication materials. This is particularly relevant in light of the expected increase in risk communication about drinking water as new information on the toxic potential of PFAS is published. This recent information is likely to affect national drinking water safety limits for PFAS [67].

In addition to improving the future international risk communication on PFAS in drinking water, the identified information, as shown in Table 1, can also be used to guide communication strategies for other emerging drinking water contaminants. Examples include communication on pharmaceuticals in drinking water [68] and the role of drinking water in the dissemination of antibiotic resistance [69].

### 4.3. Strengths and Limitations

A specific strength of this study is the interdisciplinary and participatory approach, providing different sources of information to develop the materials. By addressing both chemical and microbiological risks and involving natural and social scientists, professional stakeholders and members of the general public, we aimed to tackle professional, disciplinary and institutional boundaries. Although this process towards mutual understanding is rather challenging, it provides the opportunity to learn and benefit from one another’s knowledge and is likely to result in broadly beneficial and more socially robust risk communication.

The effectiveness of information materials is often evaluated by assessing the impact on knowledge [60]. However, choices made by consumers are often based on other factors too, irrespective of knowledge. This is also likely for knowledge about drinking water safety, as in our study we did not find a clear link between knowledge, concerns about contaminants, acceptance of norms and regulation and restrictions in water use. For this reason, this study aimed to identify the gaps between expert and lay views on risks related to emerging contaminants in drinking water and, specifically, to explore test information materials that were tailored to the information needs and that tested the effects on evaluation of information, concerns and acceptance rather than on consumers’ knowledge.

The difference in content and length of the risk messages can be considered a limitation in this study. In particular, the condition with the infographic differed considerably from the other two conditions. It not only presented a different type of information but also required more reading time and engagement from the participants. This has likely affected the results. An alternative and perhaps a more insightful approach would have been to test the infographic in comparison with other existing illustrations of the risk assessment process.

A second consideration is the participants’ representativeness. Online panels are susceptible to sampling bias. Although the invited panel of participants was a stratified random sample drawn in terms of the distribution of gender, age, education level and geographical regions in the Netherlands according to Statistics Netherlands (2018), we cannot exclude that sampling bias may still have occurred.

Finally, it should be noted that we carried out this experiment in the European context, in one country, the Netherlands. In the Netherlands, the trust in the quality and safety of drinking water is generally very high as compared with other (European) countries [52]. When using the results in other countries, the context in those countries should be accounted for. However, in view of the implementation of the revised European Drinking Water Directive ((EU) 2020/2184) and the important role given to information and risk communication, this study offers insights into possibilities to develop more effective risk communication strategies.

### 4.4. Avenues for Future Research

The experiment in this study was carried out to gain insight into the information needs and communication strategies regarding emerging chemical and microbiological drinking water contaminants, and focused specifically on the emerging chemical contaminant GenX. Chemical contaminants in drinking water primarily impose a health risk after prolonged exposure. In addition, emerging microbiological contaminants may cause acute health effects, as illustrated by a recent outbreak of gastrointestinal illness in Finland due to sapoviruses in drinking water [27]. Thus, information needs and communication strategies could be different, but it is yet unknown whether these risks are perceived differently by consumers. A similar experiment could be carried out for an emerging microbiological contaminant such as SARS-CoV-2, which, although this was not clear at first, now does not appear to be waterborne [70,71].

Furthermore, drinking water is not the only exposure route for emerging contaminants. Food may be another exposure route, and so is air, but it is unknown how consumers weigh these different exposure routes and what information needs they have on this topic. Information needs regarding emerging chemical and microbiological contaminants can also vary for different groups (e.g., the chronically ill, parents of young children, the elderly), which is an interesting avenue for future research as well.

## 5. Conclusions

Emerging chemical and microbial contaminants in drinking water resources are expected to become more of an issue in the near future. It is currently not technically viable to remove all contaminants, yet most consumers expect safe drinking water, free of contamination. This provides a new challenge for effective risk communication, i.e., how to adequately inform consumers about the presence of contaminants in drinking water while achieving or maintaining trust in drinking water safety among different consumer groups. In this study, we demonstrated that a mental models-based communication strategy, specifically tailored to consumers’ information needs, resulted in smaller increases in risk perception compared with existing strategies. The presented results can be used by drinking water companies as well as public health institutes to develop (web) texts on emerging contaminants in drinking water that do not hamper perceived drinking water safety among consumers.

## Figures and Tables

**Figure 1 ijerph-18-10615-f001:**
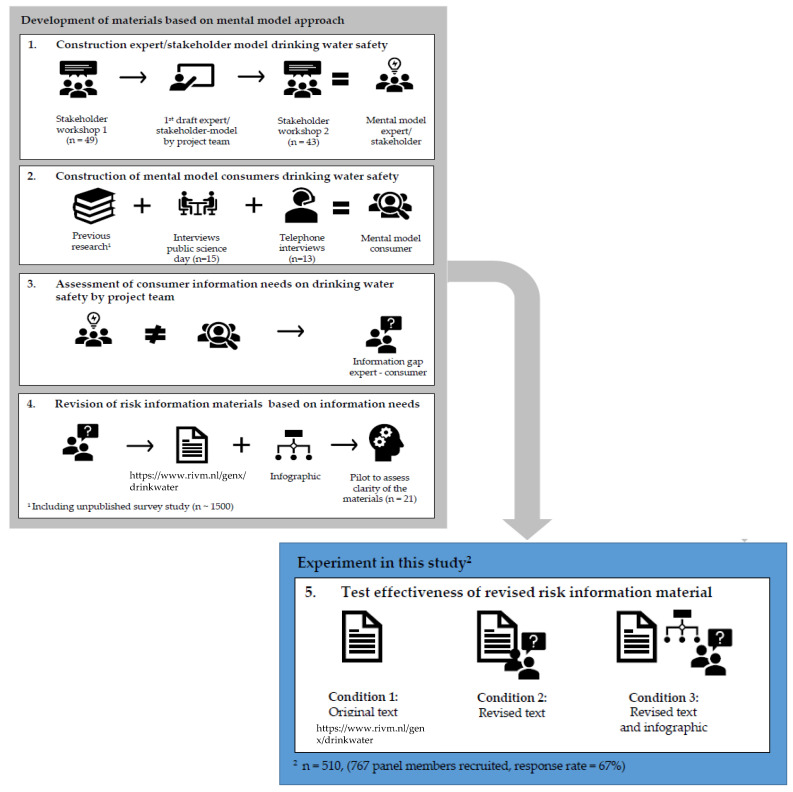
Overview of the methodology applied in this study.

**Table 1 ijerph-18-10615-t001:** Results from the comparison of the developed expert/stakeholders’ mental model and consumers’ mental model.

Lay Knowledge and Knowledge Gaps	Lay Beliefs	Lay Questions
Industry and agriculture are recognised as polluters, but transport (e.g., shipping) is not.	Increased tap water hardness is unhealthy.	*What contamination is* *found in drinking water? ^1^*
Some knowledge about chemical threats to drinking water safety (e.g., GenX, plastic), but microbiological hazards are mostly unknown.	*Drinking water treatment processes remove all* *contaminants. ^1^*	*What are the sources of chemical contamination? ^1^*
Knowledge about drinking water resources is very limited.	*If water is not free of* *contamination, it will make you sick.*	*How are risks assessed,* *and what is the basis for safety levels? ^2^*
*Responsibilities and tasks concerning drinking water safety of local, regional and national authorities and drinking water companies are unclear. ^1^*	-	*What kind and level of industrial emissions are permitted? ^1^*
-	-	*What does it mean when a contaminant is present in drinking water ‘under the (safety) limit’? ^1^*

^1^ The knowledge gaps, lay beliefs and questions in italics were addressed in the revision of the web text (https://www.rivm.nl/genx/drinkwater (last accessed on 4 October 2021)). ^2^ A supporting infographic was designed (following recommendations of Fandel, Breshears [56]) to answer the question in bold.

**Table 2 ijerph-18-10615-t002:** Demographic characteristics of the study sample.

Category	Range	n (%)	n (SD)
**Gender:**			
Male		271 (53)	
Female		239 (47)	
Age	18–89		51 (17)
**Education level:**			
High		149 (29)	
Medium		209 (41)	
Low		152 (30)	
**Drinking water regions:**			
Southwestern region ^1^		259 (52)	
Other regions		252 (48)	

^1^ In the Southwestern provinces of the Netherlands, GenX was found in the drinking water.

**Table 3 ijerph-18-10615-t003:** Pre-text evaluations.

Pre-Text Measurement	Means (SD)	
	SW-Region	Other
**Quality of drinking water** (range: 1–5)	4.5 (0.6)	4.6 (0.6)
**Knowledge about safety** (range: 6–30)	21.8 (2.9)	21.9 (2.5)
**Concerns about contaminants**		
**in drinking water** (range: 1–5):		
Carcinogenic substances ^1^	1.7 (0.9)	1.7 (0.8)
Endocrine disruptors	2.0 (1.0)	1.9 (0.9)
Micro-organisms	2.0 (0.9)	2.0 (0.8)
Calcium	2.2 (1.0)	2.2 (1.0)
Trust in risk management (range: 1–5)	3.9 (0.8)	4.1 (0.7) ^2^

^1^ These concerns are strongly correlated with concerns about endocrine disruptors (r = 0.70) and micro-organisms (r = 0.59) and moderately correlated with concerns about calcium (r = 0.30). ^2^ significant difference between regions: *p* < 0.05).

**Table 4 ijerph-18-10615-t004:** Associations between outcome variables.

	Knowledge	Trust	Evaluation of Information	Newness of Information	Concerns	Perceptions of GenX Risk	Acceptance of Norms	Restricted Water Use
**Pre-text measurement**								
Quality of drinking water	0.089 ^1^	0.479 ^2^	0.347 ^2^	−0.035	−0.287 ^2^	−0.410 ^2^	0.322 ^2^	−0.444 ^2^
Knowledge about safety		0.132 ^2^	0.242 ^2^	0.089 ^1^	−0.017	−0.136^2^	0.135 ^2^	−0.206 ^2^
Trust in risk management			0.375 ^2^	−0.053	−0.332 ^2^	−0.445^2^	0.388^2^	−0.437 ^2^
**Post-text measurement**								
Evaluation of information				0.104 ^1^	−0.307 ^2^	−0.440 ^2^	0.380 ^2^	−0.260 ^2^
Newness of information					−0.135 ^2^	−0.060	0.028	0.002
Concerns about GenX						0.694 ^2^	−0.602 ^2^	0.373 ^2^
Perceptions of GenX risk							−0.731 ^2^	0.505 ^2^
Acceptance of norms								−0.391 ^2^

^1^ Correlation is significant at the 0.05 level (2-tailed). ^2^ Correlation is significant at the 0.01 level (2-tailed).

**Table 5 ijerph-18-10615-t005:** Post-text outcomes per condition and region.

Post-text Measurement ^1^	Existing Web Text		Alternative Web Text		Alt. Text + Infographic	
			Means (SD)		Means (SD)	
	S-W-Region	Other	S-W-Region	Other	S-W-Region	Other
	*n* = 82	*n* = 86	*n* = 97	*n* = 73	*n* = 80	*n* = 92
**Evaluation of information**	3.8 (0.7)	3.9 (0.7)	4.1 (0.7)	3.9 (0.7)	4.0 (0.7)	3.7 (0.7)
**Newness of information**	2.5 (1.3)	2.4 (1.2)	2.6 (1.3)	2.4 (1.3)	2.8 (1.3)	2.4 (1.3)
**Post-text concerns:**						
Carcinogenic substances	2.3 (1.0)	2.5 (1.0)	2.2 (1.0)	2.3 (0.9)	2.3 (1.0)	2.1 (0.9)
Endocrine disruptors	2.4 (1.0)	2.3 (1.0)	2.2 (1.0)	2.3 (1.0)	2.3 (1.0)	2.2 (1.0)
Micro-organisms	2.3 (1.0)	2.1 (0.9)	2.0 (0.8)	2.3 (0.9)	2.2 (0.9)	2.1 (0.8)
Calcium	2.2 (0.9)	2.0 (1.0)	2.1 (0.8)	2.1 (1.0)	2.1 (1.0)	2.1 (1.0)
**Perceptions of GenX risk**	2.8 (0.7)	2.7 (0.6)	2.6 (0.7)	2.5 (0.7)	2.7 (0.8)	2.5 (0.7)
**Acceptance of norms**	3.2 (0.9)	3.4 (0.8)	3.3 (0.9)	3.4 (0.8)	3.3 (0.9)	3.5 (0.8)
**Restricted water use**	8.3 (3.4)	7.1 (3.0)	7.4 (3.0)	7.3 (2.8)	7.9 (3.1)	7.2 (2.9)

^1^ Scale 1–5, except for restricted water use range: 4–20.

## Data Availability

The data presented in this study are available on request from the corresponding author.

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
