# Peer review of "How to Address Consumers’ Concerns and Information Needs about Emerging Chemical and Microbial Contaminants in Drinking Water; The Case of GenX in The Netherlands"

_ijerph, 2021, doi:10.3390/ijerph182010615_

Round 1
Reviewer 1 Report
Ln#.145:
"To identify relevant consumer information needs and construct effective communication"
> missing something?
Question for future work:
Besides water intake risk evaluation I believe that it will be interesting to evaluate populations impressions on tap water freshness and potability... For that I was wondering if... In order to better classify the "flavor / potability / taste" if it is possible to make "blind tests" for people in a survey on tap water by changing water temperature, pH, and some treatment conditions like added chlorine / ozone levels, ... to really access peoples ability to actually "taste" the water and not by using their "memory impression"
Reviewer 2 Report
Introduction
Line no-35: Please insert citation instead of web address inside the text. Request to mention in the reference.
Line no-77: Missing textbox 1. Please insert the textbox.
Line no-139: Please add more information about GenX chemicals (e.g., origin, status in drinking water, guideline value, health impacts, etc).
Materials and Methods
Line no-217: Please add one section 2.2 for intervention models and clearly describe it. Discuss the sampling procedures and application of the web-text and infographic content during your research?
Line no-228: Please delete the full stop before the word "Measurements".
Result and discussion
Line no-301: Please delete the full stop before the word "Sample".
Reference
Line no-557: Please mention DOI number in the references, such as references-2, 5, 6, and 7, etc.
Line no-582: Please mention the website visited and access date in the references 17, 21, and 25, etc.
Reviewer 3 Report
This manuscript discusses the communication strategies tailored to consumers’ information needs with respect to water safety and some emerging contaminants. the results of the study showed small but relevant increases in risk perception compared with existing strategies.
I enjoyed the manuscript that is quite complete with a lot of references and clear in its format. I have no comments apart probably the length but I do not think that any part of the manuscript can be cut or reduced so I would suggest to leave it in its present form.
Round 2
Reviewer 2 Report
Dear authors, thank you for the extensive hard work and revision of your manuscript. Now, your manuscript is much better and acceptable for publication.